# Short-Chain Fatty Acid Reference Ranges in Pregnant Women from a Mediterranean Region of Northern Spain: ECLIPSES Study

**DOI:** 10.3390/nu14183798

**Published:** 2022-09-15

**Authors:** Carla Martín-Grau, Andrés Díaz-López, Estefania Aparicio, Victoria Arija

**Affiliations:** 1Clinical Chemistry and Laboratory Medicine, Institut Català de la Salut, Generalitat de Catalunya, University Hospital Joan XXIII, 43005 Tarragona, Spain; 2Department of Basic Medical Sciences, Nutrition and Mental Health Research Group (NUTRISAM), Faculty of Medicine and Health Sciences, Rovira i Virgili University (URV), 43201 Reus, Spain; 3Genetics Unit, Health Research Institute of Hospital La Fe (IIS La Fe), 46026 Valencia, Spain; 4Institut d’Investigació Sanitària Pere Virgili (IISPV), 43005 Tarragona, Spain; 5CENIT Research Group (Collaborative Group on Lifestyles, Nutrition and Smoking), Unitat de Suport a la Recerca Tarragona-Reus, Fundació Institut Universitari per a la Recerca a l’Atenció Primària de Salut Jordi Gol i Gurina (IDIAPJGol), 43202 Reus, Spain

**Keywords:** short chain fatty acid, reference ranges, pregnancy, ECLIPSES

## Abstract

Maternal short-chain fatty acids (SCFAs) play a critical role in fetal development and metabolic programming. However, an important gap in the analysis of such relationships is the lack of reference values in pregnant women. Therefore, we establish serum SCFA percentile reference ranges both early and later in pregnancy in a population from a Mediterranean region of Northern Spain. A population-based follow-up study involving 455 healthy pregnant women (mean age 30.6 ± 5.0 years) from the ECLIPSES study is conducted. Sociodemographic, obstetric, anthropometric, lifestyle, dietary variables and blood samples were collected in the first and third trimesters. Serum SCFA concentrations were measured by LC-MS/MS. The 2.5/97.5 percentiles of the reference interval for serum acetic, propionic, isobutyric, and butyric acids were 16.4/103.8 µmol/L, 2.1/5.8 µmol/L, 0.16/1.01 µmol/L and 0.32/1.67 µmol/L in the first trimester of pregnancy, respectively. In the third trimester, butyrate levels increased with most of the maternal factors and categories studied, while acetic acid and isobutyric acid decreased only in some maternal categories. Propionic acid was not affected by maternal factors. Reference ranges did not vary with maternal age, body weight, social class or diet, but decreased with smoking, high physical activity, low BMI and primiparity. This study establishes for the first-time SCFAs reference ranges in serum for women in our region in both early and late pregnancy. This information can be useful to monitor pregnancy follow-up and detect risk values.

## 1. Introduction

SCFAs including acetate acid, propionate acid and butyrate acid are important metabolites produced by anaerobic bacterial fermentation in the colon of food components, mainly dietary fiber. To a lesser extent, they are ingested directly from food or are produced endogenously through metabolic processes [1]. In addition to acting as local substrates for energy production at the intestinal mucosa, after absorption, circulating SCFAs regulate numerous metabolic pathways through the G-protein coupled receptors (GPCRs) activation [2,3], such as GPR41, GPR43, and GPR109A, which are expressed in a variety of tissues such as liver, adipose tissue, skeletal muscle, and brain. 

Some recent evidence indicates that maternal SCFAs status and their metabolism seem to play a crucial role during pregnancy. Interestingly, expression of both GPR41 and GPR43 receptors have been detected in human gestational tissues such as the uterus, myometrium, placenta, amnion, and chorion [2] which have been related to the modulation of modulating inflammatory processes involved in parturition and the functional integrity of fetal membranes [2]. Moreover, experimental evidence in pregnant rats indicated that butyric acid supplementation during early pregnancy improved embryo implantation and maintained pregnancy through enhancing maternal phospholipids metabolism and ovarian progesterone synthesis [4]. Additionally, it has been demonstrated in mice that propionate acting as GPR43 ligands influences the control of insulin levels in the embryo, and via receptors GPR41 in the fetus influence the development of the sympathetic nervous system [5]. 

Pregnancy is a period of critical physiological and physical changes. Proper adaptation to these changes will influence the health of both mother and offspring [6,7,8]. In this regard, circulating maternal SCFAs are speculated to determine the development and metabolic programming of the fetus [9,10], and therefore their imbalance may be detrimental. Supporting such effects, epidemiological studies have observed that maternal gut microbiota alteration and circulating SCFA levels are associated with a range of pregnancy complications including maternal obesity, gestational diabetes, as well as hypertensive disorders such as preeclampsia and eclampsia [9,11,12,13]; while in the offspring, associations with childhood obesity, atopic diseases, and neurological diseases, such as autism has also been reported [9,14]. These immediate but mostly long-term consequences highlight the importance to maintain adequate SCFA levels during pregnancy.

To assess the global physiological effect of SCFAs, a relatively simple and reliable method in clinical practice is the determination of serum levels of SCFAs, since they are the ones that cross the intestinal barrier and reach the blood circulation. In this sense, the Human Metabolome Database (http://www.hmdb.ca/, accessed on 24 July 2022) and Tian, Z. et al. [15], although with great variability in the data offered, has reported from few studies the mean concentration in blood samples in non-pregnant adult population (>18 years) of acetate, propionate, butyrate and isobutyrate. Concerning pregnant women, only two studies have assessed SCFAs by reporting values as mean concentrations [12,16]. However, to our knowledge, there are no reference ranges or a consensus as to what constitutes normal maternal SCFA values in pregnant women. Moreover, it is also important to know the evolution of SCFA levels during the course of pregnancy, since some studies have reported that between the first and last trimester of pregnancy, there are significant changes in SCFA-producing gut microbiota [6,17,18,19].

Considering the above-mentioned background, the need for population-based SCFA reference values throughout the course of the pregnancy and reported as percentiles will be an important contribution to the correct assessment of SCFA status both in routine clinical practice and in research. Therefore, the aim of this study was to establish serum SCFA percentile reference ranges both early and later in pregnancy in a pregnant population from a Mediterranean region of Northern Spain. In addition, the association between socioeconomic and maternal lifestyle factors and the maternal SCFAs profile has been evaluated.

## 2. Materials and Methods

### 2.1. Study Design and Participants

A population-based prospective cohort study of pregnant women was conducted from the first trimester (T1) to the third trimester (T3). Participants were pregnant women without chronic pathology, who participated in the ECLIPSES study [20,21]. This study is registered in both the ClinicalTrials.gov (identification number NCT03196882) and the European Union (EU) Clinical Trial Register (EUCTR-2012-005480-28). A detailed description of the Details of ECLIPSES have been published elsewhere [20]. The ECLIPSES trial was approved by the Clinical Research Ethics Committee of the Jordi Gol Institute for Primary Care Research (IDIAP) and the Pere Virgili Institute for Health Research (IISPV). Eligible participants were healthy adult women older than 18 years at ≤12 weeks of gestation from Catalonia, Spain. The exclusion criteria were multiple pregnancies, anemic status, taking iron supplements (>10 mg) before week 12 of gestation, hypersensitivity to egg protein, previous severe disease such as immunosuppression, or any chronic disease that could affect nutritional development (malabsorption syndrome, diabetes, cancer, hepatopathies). These exclusion criteria were applied at the beginning of the study and during gestational follow-up. Informed consent was obtained from all women included in this study. The study complies with the tenets of the Declaration of Helsinki.

Of the 793 pregnant women initially recruited, for the present analysis, all women who had data regarding serum SCFAs measurements at first (around 12 weeks of gestation) and third (around 36 weeks of gestation) trimester of pregnancy were included. Therefore, the total study sample consisted of 455 pregnant women. Participant dropout was caused by voluntary abandonment (women who have voluntarily decided to not continue with the study, *n* = 245), emergence of exclusion criteria during pregnancy (*n* = 41), miscarriage or termination of pregnancy (*n* = 14), and lost to follow-up (*n* = 36), and laboratory incident (*n* = 2). The flow of participants and serum collection is outlined in Figure 1. 

### 2.2. Data Collection

Midwives, gynecologist and nutritionists collected the participants’ medical and obstetric history (including parity (primipara and multipara) and gestational age calculated by fetal ultrasound biometry and extracted from medical records), socioeconomic information, ethnicity, education level, lifestyle habits, and anthropometric measurements was collected throughout pregnancy during the personal interview and from specific questionnaires. The socioeconomic level was calculated by using the Catalan classification of occupations (CCO-2011) [22] such as student, employed and unemployed. The education level was classified as low (primary or less), medium (high school), and high (university studies or more). Physical activity was assessed by using the International Physical Activity Questionnaire [23]. Physical activity was derived from total metabolic equivalents (METs) values based on frequency and duration of walking and moderate and vigorous-intensity activity and classified as sedentary/low (<600 METs min/week), moderate (≥600 to 2999 METs min/week), and high (≥3000 METs min/week). The Fagerström questionnaire (Fagerström_Q) [24] was used to assess smoking. The women were classified as smoker, ex-smoker and non-smokers. Anthropometric measures were height (cm) and weight (kg). Body mass index (BMI) was calculated and categorized following World Health Organization (WHO) criteria [25] as normal weight (BMI = 18.5–24.9 kg/m^2^), overweight (BMI = 25.0–29.9 kg/m^2^) and obesity (BMI ≥ 30 kg/m^2^). Total gestational weight gain (GWG) was calculated and conditioned by the initial BMI, and was categorized into insufficient, adequate or excessive GWG according to the 2009 Institute of Medicine (IOM) recommendations [26]. An adequate GWG corresponds between 11.5–16 kg for an initial normal weight, between 11.5–16 kg for an initial overweight, and between 5–9 kg for an initial obesity. Values below or above adequate GWG were considered insufficient or excessive GWG, respectively.

Usual food consumption of the women was assessed both T1 and T3 through a self-administered food frequency questionnaire (FFQ) regarding 45-food groups, previously validated in our population [27]. This questionnaire asked about the usual frequency of consumption per week or per month for each food group. Grams per day were calculated for each food group according to the standardized size and weight of a serving portion of validated questionnaire. Daily intake of energy and nutrients was calculated using the REGAL (Répertoire Général des Aliments) food composition table [28] and was complemented by the Spanish food composition table [29]. As a result, we obtained daily intake of macronutrients such as fiber and protein which were analyzed in this study in relation to SCFA since could be related to microbiota and may change during pregnancy. From this FFQ, a Spanish diet quality index (SQDI) [30] was estimated based on intake of nine food groups (fruits, vegetables, legumes, cereals, fresh fish, meat and processed meat, dairy products, olive oil, and alcohol). The resulting score ranged from 0 to 18 points, with higher values indicating a greater adherence to the SDQI, and therefore, higher diet quality. Since there are no pre-established cut-off points for the pregnant population, scores were categorized as low (0–6 points), moderate (7–10 points), and high (11–18 points) diet quality. Regarding alcohol consumption, women were classified as non-drinker and drinker of alcohol. 

### 2.3. Short-Chain Fatty Acids Measuress

During the course of the study, fasting serum samples were collected both T1 and T3 (12 ± 0.5 and 36 ± 0.4 gestational week, respectively) of gestation into 7.5 mL tubes without anticoagulant and were left without mixing for 30 min at room temperature to enable coagulation. The serum was separated by centrifugation and stored into aliquots of 500 µL at −80°C inside the BioBank until assayed. Samples were thawed at the end of the study and simultaneously assayed to minimize inter-batch variation [20].

With respect to the sample preparation [31] and quantification of the serum SCFAs (acetate, propionate, isobutyric, and butyrate acids) by liquid chromatography tandem mass spectrometry (LC-MS/MS), briefly, the method starts with a 20 µL sample mixed with internal standard mixture in methanol to prsecipitate proteins. Then, supernatants were mixed with water, o-Benzylhydroxylamine (BHA, Sigma Aldrich, St. Louis, MO, USA) and *N*-(3-Dimethylaminopropyl)-*N*′-ethylcarbodiimide (EDC, Sigma Aldrich) to obtain SCFAs derivatives. SCFAs derivatives were purified by a liquid-liquid extraction using diethyl ether and SCFAs quantification was performed by LC-MS/MS using the Ultra-high performance liquid chromatography (UHPLC) 1290 Infinity II Series coupled to a QqQ 6470 Series^®^ (Agilent Technologies Inc., Santa Clara, CA, USA). The chromatographic separation was performed with an elution gradient using a ternary mobile phase containing water, methanol and isopropanol with ammonium formate on the analytical Kinetex Polar C18 (2.6 μm 2.1 × 100 mm) column (Phenomenex, Torrance, CA, USA). The mass spectrometer operates in multiple reaction monitoring (MRM) mode and SCFAs were ionized by positive electrospray. The UHPLC-MS/MS system was controlled by the Agilent MassHunter^®^ Workstation (Agilent Technologies Inc., Santa Clara, CA, USA). The samples were analyzed in duplicate and the mean of the two values was calculated.

### 2.4. Statistical Analysis

Descriptive data were presented as mean ± SD for quantitative variables or number (%) for qualitative variables. The range, mean, and percentiles of the maternal serum concentration expressed as µmol/L of SCFAs (acetate, propionate, isobutyrate, and butyrate) during first and third trimester of pregnancy were profiled. The SCFAs reference intervals were determined following the Clinical and Laboratory Standards Institute (CLSI) C28-A3 guidelines [32] and represented the central 95% of the tested population (being the 2.5% and 97.5% confidence intervals the lower and upper limits, respectively). In this study, outlier values were identified and handled such as Martin-Grau, C et al. [33]. The means between groups (the first and third trimesters of pregnancy) were compared by the paired Student’s *t*-test. While ANOVA or independent-sample *t*-tests, as appropriate, was used to determine statistical differences in the distribution of SCFAs for intragroup comparisons. The subgroup-variable included the following prenatal characteristics: maternal age (<25, 25–29, ≥30 years), initial BMI (normal weight, overweight, obesity), gestational weight gain (insufficient, adequate, excessive), social class (low, medium, high), maternal smoking status (no, yes), parity (primiparous, multiparous), alcohol consumption (no, yes), physical activity (low, moderate, high), SQDI score (low, moderate, high), fiber (in terciles) and proteins (in terciles). Additional multivariable-adjusted analysis to investigate the independent association between maternal factors and circulating SCFA was performed (Appendix A). Results were reported as mean ± SD. All of the statistical analyses were run by the statistical software package SPSS version 25.0 for Windows (SPSS, Chicago, IL, USA). A *p*-value < 0.05 was considered statistically significant.

## 3. Results

### 3.1. Participants’ Characteristics

Table 1 shows the baseline sociodemographic and lifestyle characteristics of the pregnant women participating in the ECLIPSES study. Overall, the mean age was 30.6 ± 5.0 years old, and 23.3% of them had over 35 years old. Most of the women were Spanish (84%), 38% had a medium/high educational level, 22% were of high social class, 17% of them smoked during pregnancy, and up to 87% of them were employed. Their mean BMI initial was 24.8 ± 4.3 kg/m^2^ and about 13% of women were stratified as obese with BMI ≥  30 kg/m^2^. The mean GWG was 10.0 ± 3.6 kg. According to the IOM recommendations, 39% met, 41% fell below and 20% exceeded the criteria for GWG. The participants reported that 15.3% smoked and 14% drank at the beginning of pregnancy. The mean Physical activity was 709 ± 961 METs min/week and 56.4% of the women had a low level. The SQDI scores ranged from 4 to 17 points with the mean score being 9.7 ± 2.3.

### 3.2. Short Chain Fatty Acids in Serum of Pregnant Women

Table 2 reports the means, ranges (min/max), and percentiles (from 2.5th to 97.5th) of serum SCFAs concentrations at both T1 and T3 of pregnancy expressed as µmol/L for the entire study population. The reference interval 2.5th/97.5th percentiles for acetic, propionic, isobutyric, and butyric acids in serum were 16.4/103.8 µmol/L, 2.1/5.8 µmol/L, 0.16/1.01 µmol/L, and 0.32/1.67 µmol/L in the first trimester of pregnancy, respectively. At the third trimester, the reference interval 2.5th/97.5th percentiles of SCFAs were quantitatively similar to those observed at the beginning of pregnancy.

In both trimesters, acetic acid was present in higher concentrations in serum (T1, 49.0 ± 21.4 µmol/L; T3, 48.5 ± 18.2 µmol/L) followed by butyric acid (T1, 0.79 ± 0.33 µmol/L; T3, 0.91 ± 0.42 µmol/L) and isobutyric acid (T1, 0.47 ± 0.19 µmol/L; T3, 0.45 ± 0.24 µmol/L), while serum propionic acid concentrations were lower (T1, 3.54 ± 0.87 µmol/L; T3, 3.52 ± 1.03 µmol/L) (Table 2). From T1 to T3, there was clearly significant increase in butyrate globally (0.91 ± 0.42 µmol/L, *p* <0.001). A comparison of the mean of circulating SCFAs by sociodemographic and lifestyle factors of the pregnant women is shown in Table 3. Despite some differences, most of the maternal factors had low influence in the concentrations of SCFAs or reference ranges. In T1, only propionic acid was affected by maternal factors, the mean values decreased significantly in the subgroups of smokers’ women, and in women who practice intense physical activity. Nevertheless, levels of propionic acid at the T3 were higher in obese women and multiparity. In T3, pregnant smokers also had lower values of isobutyrate and butyrate. In addition, the high physical activity decreased isobutyrate levels in T3. 

When comparisons of SCFAs concentrations between trimesters were performed, no changes were observed for serum propionate levels throughout the course of the pregnancy according to maternal factors (Table 3). However, a significant decrease in values of acetate were observed in T3 in mothers younger than 25 years, in those women who drunk alcohol and had poorer diet quality. Related to isobutyrate levels, lower values were significantly observed in women with adequate weight gain, low social class, intense physical activity and poorer diet quality in T3. By contrast, most of the maternal factors, and in almost all of their categories, influence the increase in butyric acid levels from T1 to T3.

## 4. Discussion

To our knowledge, this is the first study that has explored the serum levels of SCFA, acetate, propionate, butyrate, and isobutyrate throughout pregnancy in a cohort of pregnant women from a Mediterranean region of Northern Spain. The current study describes the status of SCFA and the reference ranges corresponding to the 2.5th and 97.5th percentile interval at the beginning and the end of the pregnancy, and also assess the influence of maternal factors on the serum levels of SCFA, which there is scarce evidence of in the literature. The data obtained in this study will allow for greater control of SCFA serum levels during pregnancy follow-up, and detect levels outside the normal range, which may be linked to complications for the mother and her offspring [12,13].

Despite the importance of preserving a healthy state during pregnancy, there are no reliable reference ranges representing healthy pregnant women to judge the status of SCFAs, making medical/therapeutic decisions, or other physiological assessments during the gestational process. Based on the literature, we did not find any community-based study that analyses serum concentrations of SCFAs in pregnant women. Only two studies with a small sample size determined mean concentrations of SFCAs in the first trimester of gestation which mean values of SCFA differs from our results. First, Priyadarshini, M. et al. [12] have analyzed mean values for acetic acid (25.6–26.9 μmol/L), propionic acid (1.8–2.0 μmol/L) and butyric acid (0.5–0.9 μmol/L) in obese (*n* = 10) vs. normal weight (*n* = 10) pregnant women during the first trimester of pregnancy by gas chromatography. Second, Bahado-Singh, R.O. et al. [16] have reported mean values for acetic acid (28.2–22.9 μmol/L) and isobutyric acid (6.7–5.6 μmol/L) in British women with fetal aneuploidies (*n* = 30) vs. normal control cases (*n* = 114) during the first trimester of pregnancy employing nuclear magnetic resonance. The first trimester of pregnancy is described as a state of low-grade inflammation at the gut mucosal surface characterized by an increase in Ruminococcus and Faecalibacterium (butyrate-producing microorganisms with anti-inflammatory activity) [7,17,34], which is similar to that observed in healthy, non-pregnant women [17,19]. Therefore, basal fatty acid status during the first trimester should resemble those published in healthy adults. In this sense, some studies in the adult population are available in the Human Metabolome Database (HMD) (http://www.hmdb.ca/, accessed on 24 July 2022) which compiles the mean concentration of SFCAs in blood samples of adults (>18 years old) and values from Tian, Z. et al. [15]. In particular, 41.9- 69.1 μmol/L of acetic acid (*n* = 21, with NMR spectroscopy and *n* = 40, with UHPLC-MS/MS system, respectively compiled in HMD), 1.1–2.84 μmol/L of propionic acid (bibliography not available in HMD and *n* = 14, with GC-MS system [15]), 0.3–2.7 μmol/L of butyric acid (bibliography not available in HMD and *n* = 14, with GC-MS system [15]), and 2.3 μmol/L of isobutyric acid (bibliography not available in HMD). The mean values showed by these few publications from this database show that mean values for acetic (49.0 μmol/L) and butyric (0.79 μmol/L) acid are similar to our cohort of healthy pregnant women, whereas we found higher mean values for propionic (3.54 μmol/L) acid and lower mean values for isobutyric (0.47 μmol/L). The differences in these values can be attributed to several causes. One point to consider is that the methodology used in these previous studies differed from one study to another so comparisons among them are difficult. In addition, the studies discussed (http://www.hmdb.ca/, accessed on 24 July 2022 and [15,18,19]) have a very small sample size and, therefore, the presence of extreme intra-individual values can greatly modify the mean and cannot be considered as population values.

Recent research has shown that the composition of the gut microbiota changes significantly from T1 to T3 [19] and, thus, may affect SCFAs production. The highest concentration of SCFAs is found in the lumen of the colon, where they are readily absorbed by colonocytes in a concentration-dependent manner. If not metabolized by colonocytes, SCFAs are then released from the gut via the hepatic and portal venous systems [35,36]. Some authors suggested that the liver takes up propionate and butyrate, thus decreasing release of these two SCFAs into the systemic circulation. However, acetate apparently escapes hepatic metabolism to some degree and is present at a higher concentration in the peripheral circulation than either propionate or butyrate [36,37]. Moreover, acetate enters the peripheral circulation to be metabolized by peripheral tissues [38]. Related to the serum SCFAs levels, our results support the idea that the acetate is the most abundant SCFA such as the Human Metabolome Database and other authors [9]. However, no study to date has evaluated maternal SCFAs status in the third trimester of pregnancy, and therefore, their changes during pregnancy was unknown. Overall, our results showed that SCFA levels remained relatively constant over the course of pregnancy with some minor exceptions for acetic acid and isobutyric acid, while there was a large change for butyric acid. Butyric acid increased significantly with all maternal factors studied and in almost all category of each factor from early to later pregnancy. This change is important because the butyrate has been described as an anti-inflammatory agent [1,37] that increased to counteract the higher degree of inflammation that occurs in late pregnancy compared to T1 [6]. This fact could be related to the change in the intestinal microbiota during pregnancy as an increase in Actinobacteria and Proteobacteria Phyla and a reduction in Faecalibacterium has been reported, which induced more intestinal inflammation, and increased energy storage with maternal weight gain and hyperglycemia [6,18,19]. Changes in gut microbiota during T3 are very similar to those that occur in patients with certain inflammatory metabolic syndrome such as obesity or diabetes [6,12,19]. Nevertheless, despite the decline in Faecalibacterium during T3 described by some authors [6,19], a slight increase in butyrate levels (from 1.5% to 1.7%) was detected in our pregnant women. Therefore, variation in butyrate concentrations during pregnancy does not appear to be related to the described changes in the microbiota but could be influenced by other characteristics such as the amounts of microbiota present in the colon, the source of the substrate, intestinal transit time [9], host genetics, dietary composition, lifestyle factors [39]. However, in our results we observed that the increase in butyric acid levels from T1 to T3 was detected in all categories of the factors studied, so it seems that the increase in butyric acid levels is independent of these maternal factors. Further studies are needed to understand the underlying mechanisms of these changes. In contrast to butyric acid, only some environmental characteristics assessed in our sample modified acetate (younger age and alcohol consumption) and isobutyrate (low social class, adequate gestational weight gain and high physical activity) levels, decreasing them in T3.

In terms of dietary composition, many authors claim that SCFA concentrations increase when following a correct diet with high-fiber soluble products, while the production of SCFAs is reduced with the use of a fiber-restricted or high-fat diet [17,18,19]. Nevertheless, these findings are not supported by our results, nor a recent systematic review of 44 studies on the impact of dietary fiber on SCFA production, which showed that most studies (a total of 26) did not show significant differences in individual SCFA levels respected to fiber intake, while the others reported significant differences for one or only for some SCFA, but differing in the type of SCFA [40]. In our study, significant differences are only detected for butyric acid which increased in T3 compared to T1 when fiber intake was <14 g/d (1st and 2nd tercile) and when protein intake was <61 g/d (1st and 2nd tercile). Indeed, some authors have observed that the concentration of SCFAs seems to be dosage-influenced and type and structure of dietary fibers [40]. Normally, gut bacteria rely on carbohydrates and fiber for energy and use protein as an energy source when the first ones are scarce [39]. In this study, only women who maintain a moderate-high quality diet increased their butyrate levels in T3 compared to T1. It is noteworthy to consider that in our cohort, the reference values provided for serum SCFAs are independent of diet and fiber intake.

In term of lifestyle factors, women who consume alcohol, are younger, have lower social class and practice high physical activity during pregnancy, reduce acetic acid or isobutyric acid concentrations in T3 compared to T1. In general, physical activity raises Faecalibacterium (Firmicutes phyla) and modifies microbial composition [39]. In fact, athletes produce an abundance of fecal butyrate concentrations [41]. Nevertheless, an increase in butyrate concentration is observed in our women when they practice low-moderate physical activity in T3 compared to T1. It should be noted that only 4.2% of our pregnant women were highly physically active and therefore a larger number of women would be needed to establish the trend of butyric acid in these physical conditions. With regards to tobacco, smokers’ women have a tendency to lower mean values for all SCFAs compared to non-smokers and in the case of butyric acid, the decrease was significant. This difference is observed throughout the pregnancy but is most significant during T3. One of the mechanisms of absorbing SCFAs across intestinal barrier include SCFA transporters such as monocarboxylate transporter that can also use nicotinic acid as substrate [37]. Consequently, SCFAs transportation could be reduced by the presence of nicotinic acid, which acts as a metabolic inhibitor of SCFAs. Respected to BMI, the propionate is higher in obese women compared to women with normal weight. In human and experimental mice studies, the clinical effects of propionic acid manifest a reduction in fat storage, preventing insulin resistance and anti-inflammatory activity [9,42]. Priyadarshini, M. et al. [12] observed that propionate was beneficial against the development of obesity in pregnant women. 

Although small changes in mean SCFAs values between the first and third trimester of pregnancy related to maternal factors have been observed in our population, these factors as a whole have hardly affected the reference ranges. The fact of being a primiparous, together with other modifiable maternal factors such as smoking and lack of physical activity, which decrease SCFAs levels, and maternal obesity at the beginning of pregnancy, which increases them, are the only factors that modify the reference ranges described for the pregnant population in the Mediterranean area of northern Spain. Therefore, we consider that more community-based population studies are needed in different populations, due to the possible influence that these maternal characteristics may have on pregnant women in other settings.

Finally, our results can cautiously point to some of the maternal factors that negatively influence SCFAs levels and for which there are currently some plausible hypotheses about the mechanisms of action, such as excess weight in early pregnancy, smoking and intense physical activity.

One of the strengths of this study is that it is the first one to define SCFA concentrations in a large sample of healthy women at the beginning and end of pregnancy, which emphasizes the strength and validity of the proposed reference ranges in the present study for pregnant women in a Mediterranean region of northern Spain. All serum samples were collected and performed in the same research laboratory. In addition, LC-MS methodology has proven to be a robust technique for SCFA analysis that could be implemented in an automated way in laboratories and facilitate the analysis of serum samples in population-based studies with larger numbers of samples [31]. Nevertheless, we also acknowledge some limitations. SCFAs were not analyzed in the second trimester of pregnancy, and it would have been interesting to monitor the whole gestational period. Moreover, our main purpose was to establish serum SCFA reference ranges throughout pregnancy, but not studying associations with health outcomes. Further studies in larger cohorts are needed to investigate possible associations between SCFAs and adverse pregnancy outcomes.

## 5. Conclusions

The present study provides reference intervals for acetate, propionate, butyrate and isobutyrate from early to late gestation in a large sample of pregnant women from a Mediterranean region of northern Spain. SCFAs concentrations generally remained remarkably stable throughout pregnancy, except for butyrate which increased in late pregnancy, and for most of the maternal factors studied. Although SCFAs levels were hardly influenced by maternal factors, some modifiable lifestyles in early pregnancy and mainly in the third trimester of pregnancy, such as smoking, intense physical activity or obesity in early pregnancy, may modify SCFAs values in a possibly detrimental way. Further research is needed to understand the mechanism of these relationships, and to observe how SCFAs behave with these maternal factors in other populations.

## Figures and Tables

**Figure 1 nutrients-14-03798-f001:**
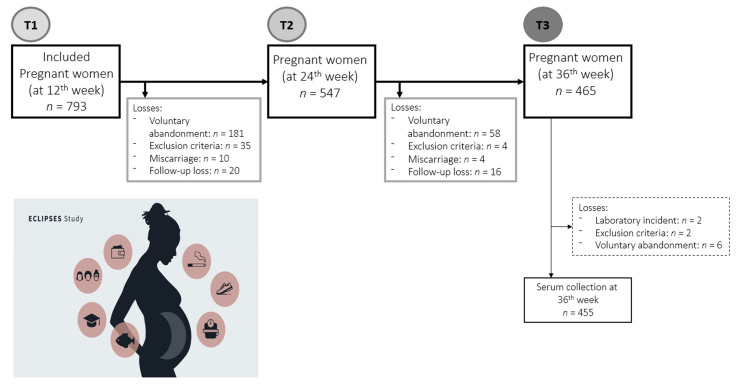
The flow of participants and serum collection during pregnancy.

**Table 1 nutrients-14-03798-t001:** Sociodemographic and lifestyle characteristics of pregnant women (*n* = 455).

Characteristics	Summary Statistics
Age (years) ^a^	30.6 ± 5.0
Country of origin, Spain (%)	84.1
Primipara (%)	37.5
Gestational age at T1 (weeks) ^a^	12 ± 0.5
Gestational age at birth (weeks) ^a^	39.8 ± 1.1
BMI (kg/m^2^) at first trimester (%)	
18.5–24.9 (normal weight)	62.2 (22.1 ± 1.7) ^a^
25.0–29.9 (overweight)	25.3 (27.3 ± 1.3) ^a^
≥30 (obesity)	12.5 (33.3 ± 2.9) ^a^
Gestational weight gain (kg) ^a^	10.8 ± 3.6
Educational level (%)	
Low (primary or less)	30.1
Medium (high school)	38.3
High (university or more)	31.6
Occupation (%)	
Student	2.4
Employed	87.1
Unemployed	10.5
Smoking status (%)	
Smoker	15.3
Non-Smoker	69.5
Ex-Smoker	15.3
Alcohol consumption (%)	14
Physical Activity (METs min/week) at first trimester (%)	
Low (<600)	56.4
Moderate (≥600–2999)	39.4
High (≥3000)	4.2
SQDI (score) at first trimester ^a^	9.7 ± 2.6

^a^ Values are expressed as a mean ± standard deviation (SD). Abbreviation: BMI, body mass index; SQDI, Spanish Diet Quality Index; METs, Metabolic equivalents.

**Table 2 nutrients-14-03798-t002:** Range, mean, and percentile distribution of maternal short chain fatty acid serum concentration (µmol/L) during pregnancy period.

			Range		Reference Interval	Percentile												
Short Chain Fatty Acid		Min	Mean ± SD	Max	Percentiles (2.5–97.5%)	2.5	5	10	20	30	40	50	60	70	80	90	95	97.5
Acetic acid (C2:0)	T1	4.37	49.0 ± 21.4	172.8	16.4–103.8	16.4	21.4	29.2	34.5	38.4	42.0	45.7	49.7	53.9	60.7	72.2	84.9	103.8
	T3	13.9	48.5 ± 18.2	130.4	23.3–108.1	23.3	26.8	30.8	35.0	38.8	42.5	45.0	48.6	52.6	59.1	68.9	89.3	108.1
Propionic acid (C3:0)	T1	1.60	3.54 ± 0.87	6.55	2.1–5.8	2.1	2.3	2.5	2.8	3.1	3.3	3.5	3.7	3.9	4.2	4.6	4.9	5.8
	T3	1.63	3.52 ± 1.03	7.81	2.1–6.5	2.1	2.2	2.4	2.8	3.0	3.1	3.3	3.5	3.8	4.1	5.0	5.6	6.5
Isobutyric acid (C4:0)	T1	0.19	0.47 ± 0.19	1.35	0.16–1.01	0.16	0.21	0.26	0.33	0.38	0.41	0.45	0.48	0.53	0.57	0.66	0.83	1.01
	T3	0.24	0.45 ± 0.24	1.83	0.14–1.19	0.14	0.18	0.24	0.28	0.32	0.36	0.40	0.44	0.49	0.56	0.68	0.92	1.19
Butyric acid (C4:0)	T1	0.33	0.79 ± 0.33	1.98	0.32–1.67	0.32	0.37	0.43	0.52	0.60	0.67	0.73	0.82	0.89	1.05	1.26	1.48	1.67
	T3	0.41	0.91 ± 0.42 ^a^	2.70	0.37–2.09	0.37	0.42	0.49	0.58	0.66	0.74	0.81	0.90	1.03	1.16	1.52	1.70	2.09

Abbreviation: SD, standard deviation; T1, First Trimester of pregnancy; T3, Third trimester of pregnancy. The significance of (^a^) is *p* < 0.05 compared with T1 and T3 as derived from Student’s *t*-tests.

**Table 3 nutrients-14-03798-t003:** Mean of maternal serum short chain fatty acids concentrations by selected sociodemographic and lifestyle characteristics of pregnant women.

	Short Chain Fatty Acids (µmol/L)											
	Acetic Acid (C2:0)			Propionic Acid (C3:0)			Isobutyric Acid (C4:0)			Butyric Acid (C4:0)		
	T1 (*n* = 454)	T3 (*n* = 454)	*p*	T1 (*n* = 449)	T3 (*n* = 449)	*p*	T1 (*n* = 450)	T3 (*n* = 450)	*p*	T1 (*n* = 457)	T3 (*n* = 457)	*p*
Maternal Factors	Mean ± SD	Mean ± SD	T1–T3	Mean ± SD	Mean ± SD	T1–T3	Mean ± SD	Mean ± SD	T1–T3	Mean ± SD	Mean ± SD	T1–T3
All	49.0 ± 21.4	48.5 ± 18.2	0.661	3.54 ± 0.87	3.52 ± 1.03	0.701	0.47 ± 0.19	0.45 ± 0.24	0.103	0.79 ± 0.33	0.91 ± 0.42	<0.001 *
Age (years)												
<25	52.6 ± 22.7	43.8 ± 12.7	0.023 *	3.52 ± 0.83	3.39 ± 0.94	0.351	0.46 ± 0.18	0.40 ± 0.17	0.101	0.81 ± 0.32	0.86 ± 0.38	0.181
25–34.9	49.4 ± 21.6	49.1 ± 18.9	0.731	3.55 ± 0.87	3.53 ± 1.05	0.872	0.47 ± 0.19	0.46 ± 0.26	0.182	0.79 ± 0.33	0.92 ± 0.44	<0.001 *
≥35	46.6 ± 20.8	49.6 ± 19.4	0.197	3.55 ± 0.91	3.54 ± 1.05	0.954	0.45 ± 0.19	0.46 ± 0.25	0.726	0.79 ± 0.35	0.87 ± 0.37	0.040 *
BMI (kg/m^2^) at T1												
18.5–24.9 (NW)	49.3 ± 22.0	47.6 ± 18.1	0.305	3.53 ± 0.87	3.41 ± 0.96	0.134	0.48 ± 0.20	0.44 ± 0.23	0.071	0.81 ± 0.34	0.90 ± 0.41	<0.001 *
25.0–29.9 (OW)	48.6 ± 19.6	50.5 ± 18.7	0.387	3.59 ± 0.92	3.61 ± 1.00	0.985	0.45 ± 0.18	0.47 ± 0.28	0.677	0.80 ± 0.32	0.87 ± 0.40	0.086
≥30 (O)	48.9 ± 25.0	47.3 ± 19.2	0.732	3.65 ± 0.82	3.86 ± 1.34 ^a^	0.243	0.46 ± 0.19	0.43 ± 0.26	0.474	0.74 ± 0.30	0.92 ± 0.45	0.015 *
Gestational weight gain (kg)												
Insufficient	49.9 ± 20.1	47.9 ± 18.2	0.411	3.54 ± 0.92	3.46 ± 1.07	0.786	0.47 ± 0.2	0.44 ± 0.24	0.331	0.81 ± 0.36	0.86 ± 0.38	0.148
adequate	48.8 ± 22.3	48.7 ± 17.2	0.937	3.64 ± 0.91	3.56 ± 1.00	0.450	0.48 ± 0.20	0.42 ± 0.21	0.011	0.77 ± 0.32	0.91 ± 0.42	0.003 *
Excessive	48.8 ± 18.7	45.7 ± 19.1	0.353	3.51 ± 0.70	3.51 ± 1.19	0.892	0.47 ± 0.17	0.43 ± 0.27	0.133	0.77 ± 0.27	0.91 ± 0.48	0.024 *
Social class												
Low	50.7 ± 19.6	48.3 ± 17.2	0.464	3.71 ± 0.82	3.76 ± 1.15	0.543	0.51 ± 0.24	0.42 ± 0.22	0.047	0.83 ± 0.34	0.93 ± 0.50	0.123
Medium	49.0 ± 20.9	48.1 ± 18.4	0.548	3.54 ± 0.86	3.50 ± 1.01	0.469	0.46 ± 0.17	0.44 ± 0.25	0.361	0.81 ± 0.33	0.91 ± 0.42	<0.001 *
High	49.7 ± 23.7	51.1 ± 19.4	0.782	3.47 ± 0.95	3.53 ± 1.04	0.764	0.47 ± 0.20	0.48 ± 0.26	0.884	0.74 ± 0.31	0.90 ± 0.38	0.004 *
Smoking during pregnancy												
No	50.0 ± 22.5	48.9 ± 18.4	0.414	3.59 ± 0.88	3.54 ± 1.01	0.440	0.47 ± 0.19	0.46 ± 0.25	0.282	0.79 ± 0.32	0.92 ± 0.42	<0.001 *
Yes	45.0 ± 13.6	47.2 ± 17.8	0.311	3.32 ± 0.71 ^a^	3.41 ± 1.11	0.333	0.44 ± 0.18	0.38 ± 0.22 ^a^	0.172	0.81 ± 0.39	0.81 ± 0.36 ^a^	0.752
Parity												
Primiparous	50.8 ± 24.6	47.2 ± 17.5	0.127	3.53 ± 0.88	3.39 ± 1.00	0.307	0.48 ± 0.20	0.45 ± 0.25	0.302	0.82 ± 0.35	0.92 ± 0.43	0.003 *
Multiparous	48.2 ± 19.0	49.6 ± 18.9	0.396	3.57 ± 0.85	3.61 ± 1.04 ^a^	0.677	0.46 ± 0.19	0.44 ± 0.24	0.254	0.78 ± 0.33	0.90 ± 0.41	<0.001 *
Alcohol consumption												
No	49.3 ± 21.5	48.7 ± 18.3	0.687	3.55 ± 0.87	3.58 ± 1.03	0.786	0.46 ± 0.19	0.44 ± 0.25	0.145	0.80 ± 0.34	0.91 ± 0.42	<0.001 *
Yes	46.4 ± 5.38	34.3 ± 3.6	**0.037**	3.90 ± 0.36	3.00 ± 0.59	0.219	0.53 ± 0.09	0.34 ± 0.08	0.117	0.77 ± 0.10	0.66 ± 0.24	0.959
Physical Activity (METs/week) at T1												
Low (<600)	48.7 ± 21.2	48.4 ± 19.0	0.716	3.64 ± 0.92	3.58 ± 1.12	0.382	0.48 ± 0.20	0.48 ± 0.27	0.757	0.79 ± 0.31	0.89 ± 0.42	<0.001 *
Moderate (600–2999)	50.6 ± 21.6	49.2 ± 18.1	0.571	3.46 ± 0.82	3.49 ± 0.92	0.623	0.45 ± 0.18	0.42 ± 0.22 ^a^	0.119	0.82 ± 0.37	0.95 ± 0.42	0.001 *
High (≥3000)	46.7 ± 18.3	46.3 ± 11.1	0.940	3.19 ± 0.53 ^a^	3.40 ± 0.93	0.401	0.48 ± 0.19	0.36 ± 0.13	0.050 *	0.74 ± 0.24	0.89 ± 0.41	0.092
SQDI (score) in T1 or T3												
Low (0–6 pts)	54.1 ± 20.3	47.6 ± 17.3	0.115	3.67 ± 0.90	3.54 ± 1.26	0.553	0.48 ± 0.19	0.46 ± 0.29	0.672	0.85 ± 0.37	0.86 ± 0.47	0.928
Moderate (7–10 pts)	49.0 ± 21.7	46.5 ± 17.5	0.239	3.59 ± 0.87	3.53 ± 1.07	0.547	0.48 ± 0.19	0.45 ± 0.23	0.237	0.81 ± 0.33	0.90 ± 0.43	0.026 *
High(11–18 pts)	47.9 ± 22.1	47.3 ± 16.4	0.770	3.48 ± 0.87	3.53 ± 1.05	0.686	0.45 ± 0.19	0.41 ± 0.21	0.053	0.76 ± 0.31	0.89 ± 0.39	0.002 *
Fiber (g/d) in T1 or T3												
t_1_ (<10 g/d)	49.0 ± 21.8	47.4 ± 15.3	0.504	3.59 ± 0.88	3.56 ±1.07	0.803	0.46 ± 0.17	0.43 ± 0.22	0.222	0.80 ± 0.33	0.90 ± 0.41	0.031 *
t_2_ (10–14 g/d)	49.3 ± 20.7	45.5 ±16.4	0.110	3.48 ± 0.87	3.39 ± 1.07	0.436	0.46 ± 0.19	0.42 ± 0.19	0.062	0.77 ± 0.32	0.85 ± 0.39	0.049 *
t_3_ (>14 g/d)	49.6 ± 22.8	47.9 ±19.0	0.534	3.60 ± 0.88	3.63 ± 1.08	0.821	0.49 ± 0.21	0.46 ± 0.27	0.310	0.83 ± 0.33	0.91 ± 0.44	0.073
Proteins (g/d) in T1 or T3												
t_1_ (<49g/d)	47.8 ± 22.1	47.7 ± 15.5	0.960	3.47 ± 0.86	3.57 ± 1.03	0.419	0.45 ± 0.19	0.43 ± 0.22	0.335	0.79 ± 0.34	0.92 ± 0.43	0.007 *
t_2_ (49–61g/d)	50.3 ± 22.4	46.9 ± 17.8	0.194	3.55 ± 0.87	3.54 ± 1.11	0.953	0.49 ± 0.20	0.44 ± 0.23	0.064	0.76 ± 0.30	0.92 ± 0.44	0.001 *
t_3_ (>61 g/d)	49.9 ± 20.7	46.3 ± 17.6	0.142	3.67 ± 0.90	3.49 ± 1.10	0.152	0.48 ± 0.19	0.44 ± 0.24	0.248	0.84 ± 0.34	0.83 ± 0.38	0.824

Values are expressed as a mean ± standard deviation (SD). Abbreviations: T, trimester; BMI, body mass index; NW, normal weight; OW, overweight; O, obesity; METs, metabolic equivalents; SQDI, Spanish Diet Quality Index in T1 or T3, depending on the SCFA assessment trimester; pts, points; t, tercile; g/d, grams per day. The significance of (*) is *p*-value < 0.05. ^a^
*p* < 0.05 compared with the first category as derived from ANOVA or Student’s *T* tests, as appropriate. *p* values for the differences in serum short chain fatty acids concentrations between trimesters (T1 vs. T3) as derived from Student’s *T* tests.

## Data Availability

Not applicable.

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
