# Peer review of "Short-Chain Fatty Acid Reference Ranges in Pregnant Women from a Mediterranean Region of Northern Spain: ECLIPSES Study"

_nutrients, 2022, doi:10.3390/nu14183798_

Round 1

Reviewer 1 Report

Review of Nutrients- 1858481

Title: Short-chain fatty acid reference ranges during pregnancy in Mediterranean population. ECLIPSES Study.               

The authors took up an interesting topic about serum SCFAs concentrations during pregnancy in Spanish population .

The content of the publication, however, has few reservations. ·       The title mistakenly suggests that the study covered the Mediterranean population from the Mediterranean basin, and in fact only concerned the Spanish population - the title should be changed. ·      

The Abstract section at line 22 it should be (in mean age 30.6±5.0 years). ·      

In the Introduction section, there is no information that the composition of the microbiota, and thus also the production of SCFAs, is mainly determined by the type of diet used. ·      

The exclusion criteria are not clear: did pregnant women who developed complications during pregnancy, e.g. gestational diabetes, pre-eclampsia, anemia, etc., also participated in this study in the third trimester? There is no flowchart that clearly defines the participation in the study at particular stages. ·      

It is surprising why serum SCFAs were not analyzed in the second trimester and why nutritional interviews were not performed in the second and third trimesters. ·      

Maternal dietary assessment during the first trimester of pregnancy is not clear - the period of time it concerned (usually FFQ refers to the last 6 or 12 months). Such a questionnaire, completed by the participants themselves in the first trimester, would then be distorted due to severe nausea and vomiting as well as reluctance to eat. In the case of pregnancy, the 24-hour nutritional interview is a much more reliable method of assessing the diet. Along with the development of pregnancy, the diet changes, which directly translates into the concentration of SCFAs in the serum. It is also necessary to repeat the nutritional interview in the third trimester to demonstrate the relationship between the diet and the serum concentration of SCFAs. ·      

In the Data collection section, the authors indicate that anthropometric measurements were performed during the first visit. This means that the body weight used to calculate the BMI was derived from the first trimester period - this is a methodological error because, for the reasons mentioned earlier, many pregnant women lose weight during this time. Pre-pregnancy body weight should be taken into account for BMI calculations. ·      

How many replicates have the individual SCFAs in the serum been analyzed? ·     

  In the table. 1 was given 39.8 weeks as a value for gestational age - this is only a value for the third trimester. The authors should also provide data for gestational age in the first trimester. ·      

In accordance with the title and the aim of the study, the authors wanted to present reformation ranges for the serum SCFAs value, however, no table was presented with the collected reference ranges for individual short-chain fatty acids in the first and third trimesters. In the Abstract section, the authors indicated only the values for the first trimester. ·      

In the Discussion section in lines 343-347, the authors of the results of the SCFAs concentration analysis from the first and third trimesters refer to the diet in fact from the pre-pregnancy period. In the Discussion, the authors should consider how the diet changes in different stages of pregnancy (e.g. increase in fiber consumption in the second trimester due to constipation, recommended increase in protein and vegetable fat consumption in the third trimester), which directly affects the composition of microbiota, and thus production of SCFAs. ·      

In the limitations of the study, the authors should also take into account the lack of nutritional interview in the third trimester. ·      

The authors made too far-reaching conclusions. First, they studied the population of Catalonia and not the entire Mediterranean population, therefore the reference range cannot be translated to the entire Mediterranean population. Secondly, the obtained results differ significantly from the results of other authors, which suggests the need to re-perform analyzes on a similar population. Third, the study report indicates that the authors had no control over the health of the study participants during the second trimester, when various pregnancy complications may have occurred requiring diet or pharmacotherapy, affecting the microbiota composition and thus the amount of SCFAs. Such a situation reduces the credibility of the obtained analysis results. ·      

The writing of the references requires corrections All the above arguments make the obtained results not reliable enough. The manuscript needs major revision.

Author Response

Dear Reviewer,

Thank you for considering our manuscript (Manuscript ID: nutrients-1858481) entitled: “Short-chain fatty acid reference ranges during pregnancy in Mediterranean population. ECLIPSES Study”.

We have added new corrections and we expect that the changes will improve your expectations.